# Analysis of Australia's Fiscal Vulnerability to Crisis

Gulasekaran Rajaguru [1,*], Safdar Ullah Khan [1] and Habib-Ur Rahman [2]

1 Bond Business School, Bond University, Gold Coast, QLD 4229, Australia; skhan@bond.edu.au
2 Faculty of Higher Education, Holmes Institute, Gold Coast, QLD 4217, Australia; HRahman@Holmes.edu.au
* Correspondence: rgulasek@bond.edu.au; Tel.: +61-755-952-049

**Abstract:** Fiscal vulnerability, like a contagion, poses a threat to financial sector stability, which can lead towards sovereign default. This study aimed to assess fiscal vulnerability to crisis by investigating the Australian economy's gross public debt, net public debt, and net financial liabilities. We used a threshold regression model and compared results with the baseline deficit–debt framework of analysis. The results of the base model suggested that the economy is fiscally sustainable, and that the primary surplus remains unaffected by increasing levels of public debt. In contrast, the threshold regression model indicated that the increasing level of debt has eroded primary surplus below the threshold level of 30.89% of public debt to GDP. These results need further investigation. Therefore, we modified our basic threshold model to capture budget deficit and surplus as a threshold in response to changes in public debt. The results from the sequential threshold regression model using the debt to GDP ratio and primary budget surplus identifying the periods of 1991, 1992, 2008, 2009, 2011 and 2019 as times of likely vulnerability to fiscal crisis. The overall results confirmed that the primary surplus remained sustainable over the estimated threshold level of public debt in all other sample periods and these findings persisted across alternative measures of public debt.

**Keywords:** Australia; fiscal vulnerability; fiscal sustainability; threshold regression; debt to GDP; primary budget surplus

## 1. Introduction

Australia's spending response to the Global Financial Crisis (GFC), bushfires, and COVID-19 provides an opportunity to study interventions from the perspective of fiscal strategy. The GFC in 2007–2009, the bushfires, and the COVID-19 pandemic in 2020 have had a substantial impact on the fiscal position of Australia, causing the budget to remain in deficit. These impacts to the budget were further enhanced by a dwindling economy that finally slipped into recession in 2019–2020. This has reduced government tax receipts, including personal income and company tax receipts, and further impacts have occurred due to increased unemployment, subdued aggregate demand, and reduced corporate profitability, which have required economic stimulus and income support to households. We argue that economic stimulus and income support for an extended period may trigger fiscal challenges. This study (a) provides early warning signals to policymakers about rollover obstacles, (b) allows policymakers to adjust policies to maintain sustainable growth, and (c) offers information on episodes of fiscal deficits which may lead to extreme fiscal stress events.

Fiscal vulnerability is a situation wherein public debt undermines primary surpluses. In this situation, a country may face a debt crisis when it is unable to pay back debt. In other words, if the government's expenditures become more than its tax revenues, and this situation continues for a prolonged period, the economy may enter a debt crisis. Many scholars have suggested that excessively expansionary fiscal policy may lead to such a crisis in many ways. For example, overwhelmingly, fiscal expansions may deplete reserves, trigger banking sector vulnerability, cause current account deficit, and result in overall unsustainability. On account of contingent liabilities, heavy deficits build up inflationary

pressures or default risks causing exchange rate fluctuations or debt rollover issues for economies. On the assets side, mounting government debt may lead to bank runs resulting in a sovereign default.

Figures 1 and 2 present historical fiscal positions, in which 2019–2020 is marked as an all-time high for the deficit. This was primarily driven by substantial payments made by the government as part of the response to COVID-19. However, much of this appears to be temporary; hence, the government receipts are expected to remain low, but payments to increase over this period.

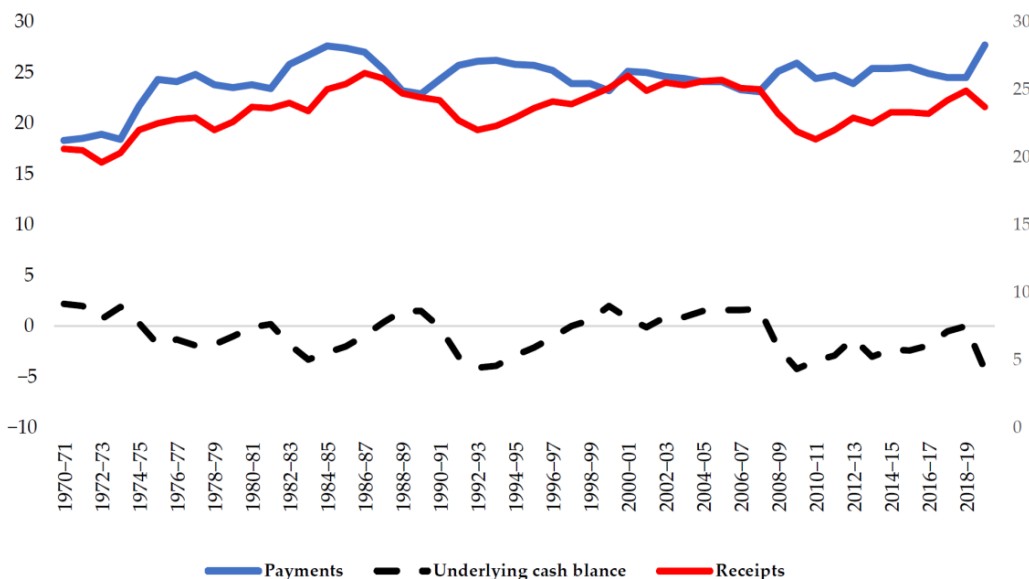

**Figure 1.** Australian Government general government sector receipts percent of GDP, payments percent of GDP, and underlying cash balance percent of GDP. Source: Australian Government, budget strategy and outlook: budget paper no. 1: 2020−2021, statement 11.

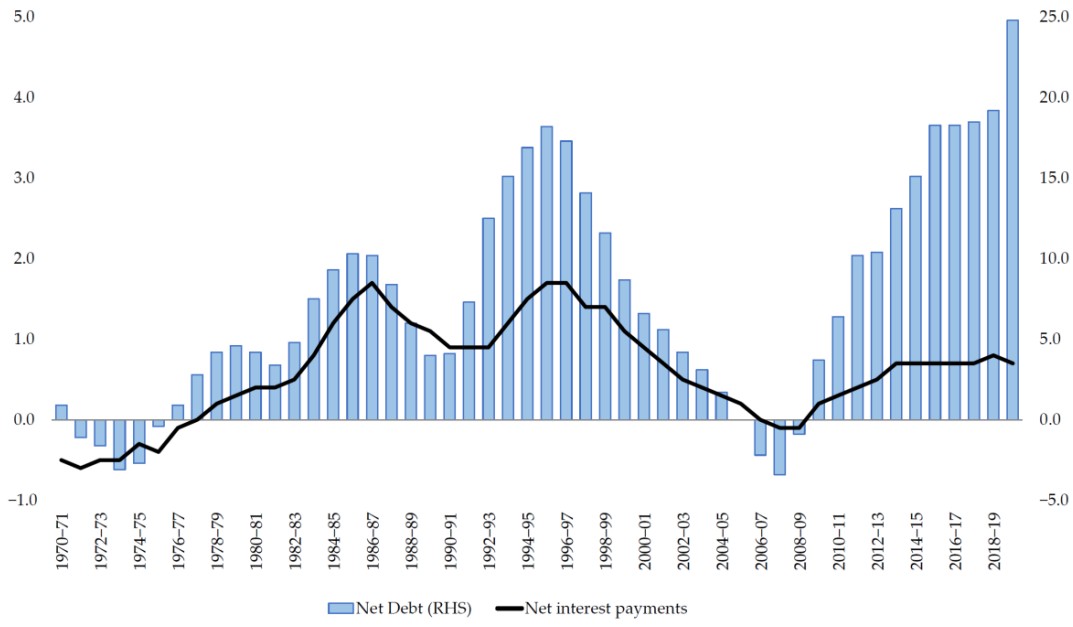

**Figure 2.** Australian Government general government sector net debt and net interest payments (percent of GDP). Notes: (a) Net debt is the sum of interest bearing liabilities less the sum of selected financial assets (cash and deposits, advances paid, and investments, loans, and placements). (b) Net interest payments are equal to the difference between interest paid and interest receipts. Source: Australian Government, budget strategy and outlook: budget paper no. 1: 2020−2021, statement 11.

Many scholars view the relationships between fiscal indicators as likely to be nonlinear. These nonlinearities suggest the presence of possible thresholds to identify fiscal sustainability or vulnerability. Therefore, we argue that estimates of benchmarks (thresholds) for fiscal indicators are vital building blocks of early warning procedures. This study is a step forward in many ways, in that we use alternative measures to investigate fiscal sustainability as determined through threshold levels of public debt and the primary budget deficit. In addition to the public debt to GDP ratio as a leading indicator of primary balances, we incorporate all offsetting accounts, including net financial liabilities, using data from the Australian general government balance sheet. The gross public debt is a commonly used fiscal measure to assess fiscal vulnerability (Bohn 1998; Herndon et al. 2014; Mauro et al. 2015; Reinhart et al. 2012). According to Valderrama (2005) gross public debt has not dealt with the public investment aspects of rising debt, which improves the government balance sheet. In particular, gross public debt does not incorporate the consolidated effect of assets, including financial and real assets. These assets enhance the net worth of a government, as revealed in Eisner and Pieper (1984). Accounting theory classifies assets as real accounts, differentiating them from nominal accounts. However, in the case of gross public debt, real and nominal accounts are not appropriately distinguished.

Consistent with accounting theory, Abelson (2012) pointed out that net public worth and net financial liabilities should be incorporated into the public debt analysis. Theoretically, the balance sheet of a public sector also measures the financial performance of an economy, because this reveals the accumulated effects of all imbalances in the budgets. Therefore, the growing level of public debt has different implications for the balance sheet of a public sector. These implications depend upon the use of public debt and can be broadly categorized into two types. First, public debt deteriorates the balance sheet if it is used to finance the budget deficit. This debt is repaid through taxation in the future. Second, public debt improves the public sector balance if the public debt is used for investment purposes. For this type of financing, it is worth noting that the government can repay the principal and the interest through different sources other than taxation. This theoretical linkage between the growing level of public debt and the balance sheet needs to be incorporated into fiscal sustainability analysis. Based on the above insights from the existing literature, we incorporated the gross public debt as a percentage of GDP ($Pd_t$), and alternative measures including the net public debt as a percentage of GDP ($Nd_t$) and net financial liabilities as a percentage of GDP ($NFL_t$), in the empirical analysis of fiscal sustainability.

The rest of the paper is organized as follows. Section 2 presents the literature review. The methodology is described in Section 3. The findings are reported and discussed in Section 4. In Section 5 we present the conclusions and implications.

## 2. Literature Review

A large number of studies in this area has focused on fiscal sustainability, ignoring fiscal vulnerability analysis. Before examining the fiscal literature, it is essential to discuss the analytical definitions of fiscal sustainability and fiscal vulnerability. Stoian (2010) provided theoretical background on the difference between fiscal vulnerability and fiscal sustainability. This difference is very small, since both concepts reflect the government's ability to generate fiscal surpluses to fulfill its financial requirements. The difference between these two concepts originates from the time horizon considered. Fiscal vulnerability is a backward-looking concept that relates to the government fiscal policy's past performance. Conversely, fiscal sustainability is a forward-looking concept that considers a long-term horizon. The earlier fiscal literature in the 1980s focused on solvency criteria or the present value budget constraint (Kremers 1989; McCallum 1984; Hamilton and Flavin 1986; Trehan and Walsh 1988; Trehan and Walsh 1991; and Wilcox 1989). Wilcox (1989), among others, revealed that the asymptotic convergence of the present value of government debt to zero indicates that the government intertemporal budget constraint holds.

Bohn (1991, 1995) criticized the fiscal literature, because prior studies examined the present value budget constraints in a certain environment. This criticism became more valid after the recent COVID-19 pandemic. Bohn (1991, 1995) suggested that an uncertain environment raises concerns about sustainable policies. Bohn (1998) developed an empirical model, based an earlier theoretical framework Bohn (1991, 1995), to analyze the behavior of U.S. public debt and deficit. On these grounds, Bohn (1998) presented an argument that the cointegration tests applied in some of the earlier studies (Trehan and Walsh 1988; Trehan and Walsh 1991; and Hakkio and Rush 1991) may have provided some misleading results. Bohn (1998) showed that the estimated positive response of primary surpluses to the debt to GDP ratio can be interpreted as the sustainability of U.S. fiscal policy.

The fiscal literature after Bohn (1998) can be classified into categories, including (1) the general framework of Bohn (Bohn 1998; Valderrama 2005; Greiner et al. 2007; Icaza 2018; Mauro et al. 2015; Ramos-Herrera and Sosvilla-Rivero 2020), (2) single threshold (Bajo-Rubio et al. 2004; Reinhart and Rogoff 2010; Legrenzi and Milas 2013; Herndon et al. 2014; Égert 2015), (3) public debt overhand (Cecchetti et al. 2011; Reinhart et al. 2012), (4) the fiscal vulnerability index (Ghezzi et al. 2010; Baldacci et al. 2011; Hayes 2011), (5) the financial net worth or balance sheet approach (Mellor 1996; Barnhill and Kopits 2004; Makin and Pearce 2016), (6) classification and regression tree (Manasse and Roubini 2009), and (7) other approaches (Afonso and Rault 2010; Chen 2014; Robinson 2002; Gruen and Sayegh 2005; Stoian et al. 2018; Hemming and Petrie 2000).

In particular, Hemming and Petrie (2000) suggested a framework for assessing four macro-fiscal aspects of vulnerability, namely incorrect specification of the initial fiscal position, sensitivity of short-term fiscal outcomes to risk, threats to long-term fiscal sustainability, and structural or institutional weakness affecting the design and implementation of the fiscal policy. However, their study failed to provide methodological aspects for their assessment. Ghezzi et al. (2010) provided a systematic global approach to develop the index of fiscal vulnerability. Baldacci et al. (2011) also developed a similar fiscal vulnerability index, which measures fiscal vulnerability continuously as a deviation from the historical 10-year country averages. Though their approach is similar to the event study methodology, it is likely to produce misleading results with the choice of 10-year averages. Hayes (2011) developed a comprehensive measure of the fiscal vulnerability index based on 16 indicators of fiscal vulnerability across 57 countries. The z-score, based on the Hayes fiscal vulnerability index, measures how far one country's vulnerability is from the cross-country average of all 57 countries. However, Hayes's approach failed to provide an absolute measure of vulnerability, and the classifications are likely to have been affected by the choice of countries. Stoian et al. (2018) developed a framework (V-L-D) to assess fiscal vulnerability for the 28 European Union countries from 1990 to 2013. The V-L-D approach classifies fiscal vulnerability into five distinct regimes, ranging from 0 (no fiscal vulnerability) to 4 (extreme fiscal vulnerability). Their results classified Greece, Portugal, Romania, the United Kingdom, Ireland, Spain, and Slovenia as the most vulnerable EU countries. Bostan et al. (2018) analyzed the vulnerabilities of public finance sustainability for Romanian data using a cointegration approach. This technique requires unit roots in at least two variables.

Bajo-Rubio et al. (2004); Reinhart and Rogoff (2010); Legrenzi and Milas (2013); Herndon et al. (2014) and Égert (2015) have used a single threshold approach to financial vulnerability. Many studies using a threshold regression model approach have used a univariate technique, such as the threshold autoregressive model (*TAR*). Estimates based on a univariate approach will be severely affected by omitted variable bias. To overcome this problem, we use the Bohn (1998) model in a threshold regression framework to identify the level of debt to GDP ratio and primary budget surplus along with other alternative measures, at which the economy is likely to be vulnerable to fiscal crisis in the case of Australia.

Some studies on Australian fiscal vulnerability (Mellor 1996; Robinson 2002; Gruen and Sayegh 2005; Di Marco et al. 2009; Makin and Pearce 2016) have suggested different measures to assess the fiscal vulnerability of the economy. The Australian government

changed its accounting of fiscal policy measures in 1996. Immediately after this change, Mellor (1996) proposed that the difference in net worth should be used to assess the fiscal vulnerability of the Australian government. They further suggest that the accrual basis of accounting should be accompanied by accrual planning and budgeting. After six years of implementing accrual accounting, Robinson (2002) reconciled the different balances of the Australian general government, including the (1) operating balance, (2) fiscal balance, and (3) cash balance. Based on this reconciliation, Robinson (2002) concluded that the net financial liabilities are the more meaningful fiscal measure among these three, even though the new measure, named the fiscal balance, is superior to the cash budget balance.

Applying descriptive analysis from 1980 to 2005, Gruen and Sayegh (2005) revealed that a sustained fiscal consolidation shifted the budget deficit (3.50% of GDP) of 1983–1984 into a budget surplus (1.75% of GDP) within five years. The Australian economy faced some severe recessions in the early 1990s, and the budget returned into deficit in 1992 (4.75% of GDP). However, the budget was again in surplus in 1997–1998. Di Marco et al. (2009) analyzed the fiscal policy of Australia from 1971 to 2008. Their analysis revealed that the gross public debt presents an incomplete picture of the public debt. They again emphasized the importance of the public sector balance sheet, and suggested that public sector assets should be incorporated into the fiscal vulnerability analysis. Comparatively, Makin and Pearce (2016) conducted a descriptive analysis using fiscal data from 1970 to 2020 (projected), and suggested three possibilities for medium-term budgetary policies. First, the government should restore fiscal solvency by moving from negative net public worth to zero. Second, the level of foreign debt should be mitigated to zero. Third, the level of net public debt should be decreased to zero.

The debt to GDP ratio for Australia is well below the Reinhart and Rogoff (2010) threshold level of 90%, but the rising level of gross public debt since 2007 is a major concern. This study mainly followed Bohn's (1998) debt-deficit model, and incorporated alternative fiscal measures guided by the current literature on fiscal sustainability. We present relevant empirical conjectures below to test the reactions of the primary budget surplus in response to public debt, and other alternative measures.

**Hypothesis 1.** *The public debt to GDP ratio has a significant effect on the primary surplus.*

Bohn (1998) demonstrates that the estimated positive response of primary surpluses to the debt to GDP ratio can be interpreted as sustainability. On the other hand, the negative relationship can be interpreted in two ways: (i) if the declining debt to GDP ratio improves the primary surplus, this is labeled as an accumulation of assets (see Mauro et al. 2015) and is still sustainable; (ii) if the increasing debt to GDP ratio lowers the primary surplus, then it is likely to indicate vulnerability to fiscal crisis. We used the above notion to describe sustainability/vulnerability by estimating the threshold levels of public debt to GDP. At the same time, the threshold levels of public debt in this exercise were not necessarily sustainable levels of debt in the case of Australia, when they exhibited a negative relationship.

**Hypothesis 2a.** *The Australian economy is sustainable at a threshold level of debt to GDP or at a threshold level of primary budget surplus in which the primary surplus increases with an increasing debt to GDP ratio.*

**Hypothesis 2b.** *The Australian economy is sustainable at a threshold level of debt to GDP or at a threshold level of primary budget surplus in which the primary surplus increases with a decreasing debt to GDP ratio.*

**Hypothesis 2c.** *The Australian economy is likely to be vulnerable to fiscal crisis at a threshold level of debt to GDP or at a threshold level of primary budget surplus in which the primary surplus decreases with an increasing debt to GDP ratio.*

This paper contributes to the literature by extending the Bohn (1998) framework in a threshold regression model to identify the debt and primary budget surplus level at which the Australian economy is either sustainable, or likely to be vulnerable to fiscal crisis.

### 3. Methodology

This section provides the empirical strategy for assessing fiscal sustainability/vulnerability in a threshold regression model framework.

Consider the empirical model proposed by Bohn (1998)

$$S_t = \beta_0 + \beta_1 L_t + \beta_2 GVAR_t + \beta_3 YVAR_t + \beta_4 D_b + \varepsilon_t \tag{1}$$

where '$S_t$' is the ratio of the primary surplus to GDP; '$L_t$' represents measures of debt, such as the gross public debt as a percentage of GDP ($Pd_t$), net public debt as a percentage of GDP ($Nd_t$) and net financial liabilities as a percentage of GDP ($NFL_t$); 'GVAR' is the level of temporary government spending; '$YVAR$' is the business cycle indicator; $D_b$ refers to break dummies, and '$\varepsilon_t$' is the error term. The debt measures were included one at a time in the model for robustness. We used Historical Table spreadsheets (available at https://www.aph.gov.au/About_Parliament/Parliamentary_Departments/ Parliamentary_Library/pubs/rp/BudgetReview202021/Fiscal; accessed on 13 June 2021), sourced from the Australian Government Final Budget Outcomes, the Australian Bureau of Statistics (ABS), the Australian Office of Financial Management, and the Australian Government Consolidated Financial Statements.

Non-debt determinants of the primary surplus (*GVAR* and *YVAR*) capture unusual variations in government spending and output, respectively. We calculated the *GVAR* and YVAR by following Barro (1986), and Mauro et al. (2015):

$$GVAR_t = (G_t - G_t^{tr})/Y_t, \ YVAR_t = (1 - (Y_t^{tr}/Y_t))(G_t^{tr}/Y_t) \tag{2}$$

where '$G_t$' is the government expenditures, '$G_t^{tr}$' is the trend in government expenditures, '$Y_t$' is the GDP, and '$Y_t^{tr}$' is the trend in GDP. From the above model (Equation (1)), (i) the significant positive value of β1 indicates that the gross public debt to GPD, net public debt to GDP, and net financial liabilities to GDP are sustainable; (ii) the significant negative value of β1 with declining debt to GDP ratio, along with an increasing surplus (i.e., if the debt to GDP declines from the period $t - 1$ to t then the primary surplus will increase from $t - 1$ to $t$), indicates sustainability; and (iii) the significant negative value of β1 with increasing debt to GDP ratio, along with a declining surplus (i.e., if the debt to GDP increasers from the period $t - 1$ to t then the primary surplus will decrease from the period $t - 1$ to $t$), indicates that the economy is likely to be vulnerable to fiscal crisis.

In order to examine the time series property of the key variables in the presence of structural breaks, the breakpoint unit root test, and three conventional unit root tests (Augmented Dickey-Fuller, Phillips-Perron and Kwiatkowski-Phillips-Schmidt-Shin), were used. The breakpoints identified through the breakpoint unit roots test were subsequently introduced as break dummies ($D_b$) in the model.

The linear model proposed by Bohn (1998) assumes that the model parameters are stable across the sample period. To overcome this problem, Bohn (1998) estimated the model at the various subsamples and assumed that the model parameters were stable within the subsample period. However, the model parameters could be unstable within the chosen subsample periods, and hence the results could be misleading. On the other hand, Mauro et al. (2015) and Rahman (2018) used the rolling window approach to identify fiscal vulnerability. The findings from Bohn (1998) and Mauro et al. (2015) indicated whether a country was vulnerable to a fiscal crisis or not. However, they did not provide the optimal debt level at which a country is likely to become vulnerable to crisis. In order to examine the fiscal vulnerability level of public debt to GDP and net financial liabilities, we extended the analysis to the threshold regression model, using specifications represented by Equation (1). The model supposes that economic time series can be modelled as belonging to a number of

distinct regimes, where the regimes are characterized by different conditional distributions of the process. Accordingly, it allows for flexibility in model parameters through regime-switching behavior. In the present case, the model assumed that parameters change once a series enters a different regime. The parameters in Equation (1) depend on the country-specific debt to GDP ratio. This approach eliminates the problem of assuming a single threshold regardless of the size of the economy. The framework of the threshold regression model is outlined below.

Consider an $m - 1$ threshold and the implied m-regime representation of ergodic stationary processes as follows:

$$
S_t = \begin{cases}
\beta_1 X_t' + \delta_1 L_t + \varepsilon_t, & \text{if } L_t \leq r_1 \\
\beta_2 X_t' + \delta_2 L_t + \varepsilon_t, & \text{if } r_1 < L_t \leq < r_2 \\
\beta_3 X_t' + \delta_3 L_t + \varepsilon_t, & \text{if } r_2 < L_t \leq < r_3 \\
\quad \vdots \\
\beta_m X_t' + \delta_m L_t + \varepsilon_t, & \text{if } r_{m-1} < L_t \leq < r_m
\end{cases}
\tag{3}
$$

where $L_t$ represents all three measures, including gross public debt, net public debt, and net financial liabilities (percentage of GDP). The delay parameters, denoted by $L_t$, and the thresholds $r_1, r_2, \ldots r_{m-1}$ are the parameters that yield the nonlinear structure of the model. Using the sequential procedure developed by Strikholm and Teräsvirta (2006), each model has two regimes for all of the specifications. Moreover, threshold regression was performed with an ordinary coefficient covariance matrix, using the threshold specification of Bai-Perron tests of $L + 1$ vs. $L$ to sequentially determine the threshold at a 15% trimming percentage. Hence, the model is as follows:

$$
S_t = \beta_1 X_t' + \delta_1 L_t + \varepsilon_t \qquad if \ Pd_t < r_1
\tag{4}
$$

$$
S_t = \beta_2 X_t' + \delta_2 L_t + \varepsilon_t \qquad if \ Pd_t \geq r_1
\tag{5}
$$

where $S_t$ is the ratio of the primary surplus to GDP, $L_t$ represents all three measures, including gross public debt, net public debt, and net financial liabilities (percentage of GDP), '$X_t$' is the vector of control variables (*GVAR*, *YVAR*, and $D_b$), and $r_1$ is the threshold values of public debt to GDP. Since the threshold is a piecewise and locally linear model, ordinary least squares (OLS) can be used to estimate Equations (4) to (5) as long as the threshold parameters are known. Hansen (2000) notes that the OLS estimator is also the maximum likelihood estimator when $\epsilon$ is iid and $r_1$ is known.

## 4. Results and Discussion

The unit test results based on Augmented Dickey-Fuller, Phillips-Perron and Kwiatkowski-Phillips-Schmidt-Shin, and Breakpoint unit test results are reported in Table 1. The results show that all variables except the debt measures are stationary in levels. The debt measures were found to be I(1) and, hence, this was used in the first-difference form in the subsequent analysis.

First, we estimated Equation (1) for the full sample from 2016 to 2020 using a conventional measure of debt, based on Bohn's (1998) methodology. The results are reported in panel I of Table 2. The results show that the gross public debt to GDP coefficient ($\beta_1 = -0.08$) is negative and insignificant when using the full sample data from 1990 to 2020. This lends support against hypothesis 1 when we consider the full-sample in a linear model, and it further indicates that the primary budget surplus does not respond to the public debt to GDP ratio—the Australian economy is not vulnerable to fiscal crisis. We posit that the results are likely to be dominated by non-vulnerable periods. Therefore, we extended the analysis to threshold regression using specifications from Equations (4) and (5), to overcome the potentially spurious results above. Panel II of Table 2 presents the results of the threshold regression model. The optimal number of thresholds was found to be one, indicating that the model exhibit two regimes. The delay (threshold) parameter

was found to be 30.89% of gross public debt to GDP. The coefficient of gross public debt to GDP below and above the threshold levels of 30.89% were −0.27 (significant) and −0.12 (insignificant), respectively. The gross public debt was statistically insignificant using a 5% level of significance when gross public debt to GDP is up to the level of 30.89%. The single most striking observation to emerge from this analysis is that the gross public debt is insignificant when the ratio is more than 30.89%, and, subsequently, the economy remains fiscally not vulnerable.

**Table 1.** Unit root test results.

|  | **ADF** | **PP** | **KPSS** | **Break** |
|---|---|---|---|---|
| S | −2.86 *** | −2.37 ** | 0.30 | −5.08 *** |
| Dp | −1.15 | −1.05 | 0.47 * | −2.19 |
| ΔDp | −3.48 *** | −3.58 *** | 0.17 | −4.72 *** |
| Nd | −1.54 | −0.88 | 0.35 * | −3.05 |
| ΔNd | −2.62 ** | −2.69 *** | 0.14 | −4.59 ** |
| NFL | −1.32 | −1.32 | 0.52 ** | −2.76 |
| ΔNFL | −3.47 ** | −3.46 ** | 0.30 | −5.19 *** |
| GVAR | −3.51 ** | −3.27 ** | 0.09 | −4.85 ** |
| YVAR | −4.40 *** | −4.47 *** | 0.06 | −4.93 ** |

Note: *, ** and *** denote rejection at the 10%, 5% and 1% level of significance.

**Table 2.** Australia's fiscal sustainability and threshold using conventional and alternate measures.

| Approach | Sample | Threshold Level | Constant | ΔPd | ΔNd | ΔNFL | GVAR | YVAR | $R^2$ | DW |
|---|---|---|---|---|---|---|---|---|---|---|
| Conventional Measures of Debt | Panel I 1960–2019 |  | 0.41 (0.99) | −0.08 (0.06) |  |  | −0.87 *** (0.12) | −0.35 *** (0.08) | 0.86 | 1.79 |
|  | Panel II 1960–2019 | Pd < 30.89 | 0.45 *** (0.13) | −0.27 *** (0.07) |  |  | −0.61 *** (0.11) | −0.16 * (0.08) | 0.89 | 1.92 |
|  |  | Pd ≥ 30.89 | 1.16 *** (0.25) | −0.12 (0.09) |  |  | −0.51 ** (0.25) | −0.08 (0.09) |  |  |
|  | Panel III 1960–2019 | S < 0.6 | −1.12 *** (0.36) | −0.42 *** (0.09) |  |  | 0.29 (0.25) | 0.08 (0.08) | 0.85 | 1.93 |
|  |  | S ≥ 0.6 | 1.80 *** (0.17) | 0.007 (0.07) |  |  | −0.45 *** (0.14) | −0.08 (0.09) |  |  |
| Alternate Measures of Debt | Panel IV 1970–2019 |  | 0.29 |  | −0.40 *** (0.072) |  | −0.58 *** (0.14) | −0.24 *** (0.096) | 0.91 | 1.94 |
|  | Panel V 1970–2019 | Pd < 30.89 | 0.48 (0.55) |  | −0.44 *** (0.07) |  | −0.55 *** (0.14) | −0.29 ** (0.11) | 0.94 | 1.89 |
|  |  | Pd ≥ 30.89 | 0.09 (0.79) |  | −0.14 (0.30) |  | −1.04 (0.64) | −0.36 (0.22) |  |  |
|  | Panel VI 1970–2019 | S < 0.6 | −1.09 *** (0.32) |  | −0.55 *** (0.11) |  | 0.73 *** (0.23) | 0.24 *** (0.08) | 0.90 | 1.75 |
|  |  | S ≥ 0.6 | 1.20 *** (0.23) |  | 0.68 (0.75) |  | −0.13 (0.15) | 0.02 (0.09) |  |  |
|  | Panel VII 1990–2019 |  | −0.027 (1.67) |  |  | −0.08 * (0.04) | −1.49 *** (0.189) | −0.58 *** (0.13) | 0.93 | 1.68 |
|  | Panel VIII 1990–2019 | Pd < 30.89 | 0.23 (1.49)) |  |  | −0.12 *** (0.03) | −1.53 *** (0.18) | −0.81 ** (0.32) | 0.95 | 1.65 |
|  |  | Pd ≥ 30.89 | 0.09 (13.09) |  |  | 0.014 (0.13) | −1.43 ** (0.55) | −0.52 * (0.94) |  |  |
|  | Panel IX 1990–2019 | S < 0.6 | −2.08 *** (0.09 |  |  | −0.07 * (0.04) | −1.03 ** (0.37) | −0.43 *** (0.13) | 0.96 | 1.65 |
|  |  | S ≥ 0.6 | 2.06 *** (0.38) |  |  | 0.09 (0.32) | −0.29 (0.69) | −0.18 (0.78) |  |  |

Note: Pd indicates the gross public debt to GDP ratio, which is used as the conventional approach. In contrast, Nd and NFL indicate the net financial liabilities to GDP and net public debt to GDP, which are used as the alternate approaches to fiscal sustainability analysis. Following Mellor (1996), changes in the net debt and financial liabilities are used in the analysis. *** indicates significance at 1%, ** indicates significance at 5% and * indicates significance at 10%.

The results pose a challenge in that public debt below the threshold level has a significant negative relationship with the primary budget balance, but is insignificant when the debt to GDP is above 30.89%—this may be contrary to common perceptions. In

particular, the negative and significant results can further be classified into sustainable or vulnerable periods: (i) the primary surplus was found to be declining with an increasing debt to GDP ratio in the years 1991, 1992, 2008, 2009, and 2011, which is in support of Hypothesis 2c that suggests that the economy is likely to be vulnerable, and (ii) the primary surplus was found to be increasing due to a declining debt to GDP ratio in all other periods, which is in support of Hypothesis 2b, when the economy is sustainable during these periods. The most disturbing findings from the threshold regression model was that the coefficient of debt to GDP is insignificant when the debt to GDP is above 30.89%. The insignificant results could be due to insufficient degrees of freedom, as there were only 15 observations above 30.89% debt to GDP.

In turn, we investigated further by considering primary surplus to GDP as a threshold variable, to analyze whether the economy is likely to be vulnerable when the debt to GDP ratio is above 30.89%. Interestingly, the estimated threshold level of the primary surplus was found to be 0.6%, hence the model split the sample into two subsamples: (i) a primary surplus below 0.6% tended to be close to a balanced budget, as the threshold value 0.6% was close to 0, and (ii) an above balanced budget (i.e., greater than or equal to 0.6%). In Table 2, Panel III presents the results showing that the public debt, indeed, worsens the primary deficit ($\beta_1 = -0.42$, significant) during times when the Australian economy suffers persistent deficit episodes (such as 1975–1978, 1982–1985, 1991–1994, 2008–2019). The above findings lend support towards hypothesis 1. Moreover, we found the evidence for hypothesis 2c (that increasing public debt leads to a decreased primary balance when primary surplus is less than 0.6%). The corresponding years were 1991, 1992, 2008, 2009, 2011, and 2019. The Australian economy was likely to be vulnerable to fiscal crisis during these periods. All other periods of primary budget deficit were in favor of hypothesis 2b; these periods are considered to be sustainable, even though they experienced a primary budget deficit. However, the primary surplus did not respond to increasing levels of public debt ($\beta_1 = 0.007$, insignificant) in the subsample of sustained budget surpluses. These results reflect the need for Australian fiscal authorities to be sensitive to primary imbalances, and always be prepared for fiscal consolidation when a budget deficit arises. Our results above are further in line with historical underlying cash balances, which appeared in deficit when responding to recessions or other negative supply shocks (see Appendix A Table A1).

We note that these results are in contrast to the single optimal level of 86% (Cecchetti et al. 2011) and 90% (Reinhart and Rogoff 2010), which have been highly discredited based on (a) coding errors; (b) exclusion of available data in selective cases; (c) inappropriate weighting (Herndon et al. 2014); and (d) ignoring country-specific circumstances. These results provide important insights for policymakers, suggesting that the current level of public debt requires caution.

Based on the criticism of Eisner and Pieper (1984) and Soos (2016), it is important to review the results for Australian general government public debt after incorporating offsetting accounts. As can be seen from panel IV of Table 2, the debt coefficient of the net public debt ($\beta_1 = -0.40$) is highly significant, which lends support to hypothesis 1. Furthermore, we have estimated the piecewise regression model using the previously established threshold value of 30.89% of gross public debt to GDP. Based on this threshold value, the piecewise regression results (panel V in Table 2) indicate that the coefficient for net public debt as a percentage of GDP is negative and significant when the gross public debt to GDP is below 30.89%. On the other hand, it is insignificant when gross public debt to GDP is above 30.89%. Equivalently, the threshold level of the net debt to GDP was found to be 13.10%. The results based on the Nd using the balanced budget threshold are consistent with pd (panel VI of Table 2).

Panel VII of Table 2 presents the results of another alternative approach, using the change in net financial liabilities of the Australian general government. The value of the NFL coefficient ($\beta_1 = -0.08$) is significant, which reveals that the Australian general government net financial liabilities, including unfunded superannuation liabilities, are not

sustainable. Similar to the previous case, we have estimated the piecewise regression model using the threshold value of 30.89% of gross public debt to GDP. Based on this threshold value, the piecewise regression results indicate that the coefficient of NFL is negative and significant when the gross public debt to GDP is below 30.89%. On the other hand, it is found to be insignificant when gross public debt to GDP is above 30.89%. Equivalently, the threshold level of NFL to GDP was found to be 23.50%. The results based on the NFL using a balanced budget threshold are also consistent with pd (panel IX of Table 2). In summary, the Australian economy is high on the sustainability scale, and the results highlight problematic episodes of fiscal vulnerability in the past, including the year 2019. The results based on the alternative measures of debt are consistent with the traditional measure of debt. The threshold regression model is in support of Hypotheses 1, 2b, and 2c.

## 5. Conclusions

This study assessed the fiscal sustainability and vulnerability of the Australian economy using gross public debt and alternative fiscal measures including net public debt and net financial liabilities. We contribute to the literature by extending Bohn's (1998) setup to threshold models from 1960 to 2019. The existing literature has widely used gross public debt to GDP ratio to identify the economy's vulnerability to a fiscal crisis. These conventional tools ignore the appropriate classification of rising debt, which improves the government balance sheet. We incorporated the above aspect by applying net debt and net financial liabilities of the Australian government, which additionally included unfunded superannuation liabilities. The results based on the sequential threshold regression model using the debt to GDP ratio and primary budget surplus identifies the periods of 1991, 1992, 2008, 2009, 2011, and 2019 as likely to be vulnerable to fiscal crisis. The results of all measures indicates that the Australian economy is likely to be sustainable even beyond the threshold value of 30.89% of gross public debt to GDP. The threshold regression model is in support of Hypotheses 1, 2b, and 2c. These unique results can be partly attributed to unutilized assets accumulated over a period of time. However, we suggest fiscal consolidation against anticipated vulnerability episodes, as have been observed historically in Australia. Recently, Inchauspe (2021) suggested fiscal expansions subject to increasing taxes, to offset the fiscal expansion required to achieve higher growth in the aftermath of the COVID-19 shock.

There is no evidence in the literature regarding a safe level of public debt for countries to follow. Nevertheless, budget constraint is commonly considered to cover the current level of public debt with future primary surpluses. Therefore, a higher debt ratio puts more burden on primary balances. Further, a higher debt ratio also implies a history of fiscal imbalances, and it is more difficult to achieve the balanced budget in years identified as fiscally vulnerable. There are some studies, mostly from a pool of emerging economies, which have attempted to estimate debt ratios to investigate the debt tolerance of that pool of economies. For example, Reinhart et al. (2003) found cases of countries with the highest debt intolerance, where foreign debt level is just 15% of GDP Daniel et al. (2003) find the majority of countries at high risk of intolerance have a public debt below 60% of GDP, and others have observed debt ratios below 40% of GDP.

Regarding advanced or developed economies there is no example of a default, hence the literature is silent on fiscal sustainability and the safe level of the debt to GDP ratio in connection with primary surpluses. However, some studies have attempted to investigate a safe level of debt from the perspective of its effect on economic growth. In this regard, studies have found that public debt beyond its critical threshold level affects economic growth negatively—hence, public debt and economic growth show a nonlinear relationship.

In our study, following the above literature on public debt and growth, we attempted to find the critical threshold level of public debt in relation to primary surpluses by recognizing the non-linear relationship between the two. We found that public debt grows faster with depleting primary surpluses up to the level of the estimated threshold, i.e., 30.89% of the GDP. The threshold level of public debt to GDP for Australia appears small

compared with many other advanced economies currently, which is perhaps due to Australia's unannounced balanced budget policy. This is coupled with the fact that Australia has not been observed to have a high ratio of public debt to GDP since 1970—the highest value was recorded as 47.47% in 2019. It is also relevant to note that the Australian debt to GDP ratio remains historically low as compared with the G8 economies. Australia perhaps prefers a balanced budget approach, therefore even low levels of public debt to GDP may appear significant in response to diminishing levels of primary surpluses. We admit that the Australian economy may have the capacity to have a higher ratio of public debt to GDP, but to estimate the potential capacity of public debt to GDP remains beyond the scope of this paper. However, vulnerability to fiscal crisis can be combated through fiscal consolidation. Future research will incorporate the role of fiscal consolidation in financial sector stability.

**Author Contributions:** Conceptualization, G.R., S.U.K. and H.-U.R.; methodology, G.R.; software, G.R.; validation, G.R. and H.-U.R.; formal analysis, G.R. and S.U.K.; investigation, G.R. and S.U.K.; resources, H.-U.R.; data curation, H.-U.R.; writing—original draft preparation, G.R., S.U.K. and H.-U.R.; writing—review and editing, G.R. and S.U.K.; visualization, G.R. and S.U.K.; supervision, G.R. and S.U.K.; project administration, G.R. All authors have read and agreed to the published version of the manuscript.

**Funding:** This research received no external funding.

**Institutional Review Board Statement:** Not applicable.

**Informed Consent Statement:** Not applicable.

**Data Availability Statement:** All data is from the Australian Bureau of Statistics (ABS) and International Monetary Fund (IMF) database.

**Acknowledgments:** We are grateful to anonymous referees for their insightful comments. We acknowledge feedback and comments offered by the audience of a seminar at the Bond Business School and the Australian Conference of Economists (2018). We have also benefited from the comments and suggestions received from Arthur Goldsmith, Washington and Lee University, USA, Omar Farooq Saqib, State Bank of Pakistan (central bank), Saeed Ahmad, International Monetary Fund (IMF), USA, Susannah Stearman and Alexandra Bec, City of Gold Coast, Australia. Views expressed here are those of authors and not their affiliated intuitions.

**Conflicts of Interest:** The authors declare no conflict of interest.

## Appendix A

**Table A1.** Australian Underlying Cash Balance (per cent of GDP).

| Fiscal Years | UCB % of GDP | Comments |
|---|---|---|
| 1975–1976 to 1986–1987 except 81–82 | −1.08 | In response to early 1980s recession. |
| 1987–1988 to 1989–1990 | 1.13 | Trilogy of commitment. |
| 1990–1991 to 1996–1997 | −2.46 | In response to early 1990s recession. |
| 1997–1998 | 0.00 | The second half of the 1990s repeated the last decade's experience with the budget returning to surplus in 1997–1998. During both decades, the tightening was owed more to discretionary fiscal tightening than the automatic stabilisers' operation. |
| 1998–1999 to 2007–2008 except 2001–2002 | 2.40 | Budget surplus until 2007–2008. |
| 2008–2009 to 2017–2018 | −2.39 | Again, this surplus was interrupted by the GFC and the underlying cash balance remained in deficits until 2017–2018. |

**Table A1.** *Cont.*

| Fiscal Years | UCB % of GDP | Comments |
|---|---|---|
| 2018–2019 | 0.00 | In 2018–2019, the Australian government returned the budget to balance (underlying cash balance of zero per cent of GDP) for the first time in eleven years. |
| 2019–2020 | −4.30 | The recent shutting of businesses and closing state and international borders due to the COVID-19 pandemic has substantially changed the Australian fiscal position. Therefore, it is vital to revise the medium-term budgetary outlook in the 2020–2021 budget. Surprisingly, the revised fiscal projections reveal that the budget will remain in deficit for at least the next decade, and the government debt will increase substantially. |

Note. The underlying cash balance (% of GDP) data is extracted from "Table 1: Australian Government general government sector receipts, payments, net Future Fund earnings and underlying cash balance" of the Historical Australian Government Data. The values are the average of UCB (% of GDP) during the period given in the column "Fiscal Years".

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
