# Peer review of "Analysis of Australia’s Fiscal Vulnerability to Crisis"

_jrfm, doi:10.3390/jrfm14070297_

Round 1

Reviewer 1 Report

Analysis of Australia’s Fiscal Vulnerability to Crisis

This paper is assessing Australia’s fiscal vulnerability by resorting to a fiscal reaction function, respectively it studies the response of the primary balance to debt to GDP ratio. Given that this relationship is likely to be non-linear the authors use a threshold regression model. The results point to a negative response of the primary balance to public debt when the threshold of 23.41% is exceeded and the authors conclude that this is a sign of vulnerability to a debt crisis.

In general, the paper is coherent and the authors present clearly their research. The econometric work seems to be well executed.  I would have one question though related to very low level for the coefficients for GVAR, YVAR.  Perhaps the authors could comment a bit more on that aspect.

I feel that the authors should better clarify the differences between fiscal vulnerability and fiscal sustainability. One could argue that a negative coefficient in the fiscal reaction function lead rather to sustainability issues. Also, the authors seem to link fiscal vulnerability to a debt crisis. A good review of the concept of fiscal vulnerability can be found in Stoian et. al (2018), an article which is already cited on this paper.

Also, I feel that the conclusions are rather strong. The authors conclude that a negative coefficient of the primary balance when debt exceeds 23.41% is equivalent to a high vulnerability of a debt crisis. Is that really plausible for a developed economy such as Australia? There are many authors who claim that developed economies can sustain much larger level of debt. The authors could either bring additional strong arguments to their statement or perhaps change the conclusion. The estimates seem to suggest that indeed fiscal policy becomes more vulnerable after this threshold but these results can be presented also in an alternative way.

Author Response

Response to Reviewer 1, attached

Reviewer 2 Report

The topic of the manuscript is interesting and it is justified to be presented in the high-quality Journal. The Authors conducted interesting and significant empirical research to analyze fiscal vulnerability to crisis, based on the Australian case. The abstract was appropriately prepared. Aim of the paper is also correct. All these aspects have to assess positively.

However, it was identified some significant imperfections that do not allow to accept the paper and proceed to the further publication process. They are as follows:

  • Almost lack of literature review. Authors present something like literature studies in the “Introduction” section but it is still insufficient. It should be a broad international literature review, that they present whether the similar research were conducted previously. What were their results? Whether someone previously used the same methodology or maybe it is innovative, developed by the Authors? What is innovative in the research? Where is the research gap? In general, what literature says about the undertaken topic and how the Authors would like to extend it.
  • There are also no: research hypothesis/hypotheses that are verified during the research, research questions, information about the research/paper contribution to the literature and general practise, what are the main limitation of the research. It has to be supplemented indisputably.
  • In high quality papers we usually use the word “methodology” rather than “empirical strategy”. From the Reviewer point of view it should be corrected. Moreover, the Authors should provide the research methods used during the research.
  • Finally, the last section “Conclusions” should be also supplemented by the information what are the basis point for the further in-depth research, how the Authors may continue the research and whether they confirm or reject the adopted research hypothesis/hypotheses.

Nevertheless, as it was mentioned at the beginning the research are important and interesting that is why it is suggested for the Authors to make significant improvements and re-submit the manuscript to the Journal. In the current form of the paper, the review can not be positive.  

Author Response

Response to Reviewer 2, attached

Reviewer 3 Report

I read carefully the paper entitled "Analysis of Australia’s Fiscal Vulnerability to Crisis".

The topic of the research is very important.

We have discovered many scientifically valuable elements.

However, I have some suggestions to make.

For example, the Introduction is too extensive in relation to the other parts of the article.  

I suggest writing the Introduction separately, followed by Literature Review.

Source - Figure 1 ???????

I recommend the following to the authors to better identify the elements of their own scientific contribution.

In particular, to specify which is the part through which the paper brings superior elements in relation to other researchers.

I noticed that relatively old / outdated bibliographic works are used. The bibliography should be extended with some papers published in prestigious WoS indexed journals (2020-2021).

Author Response

Response to reviewer 3, attached

Round 2

Reviewer 2 Report

The Authors made a huge work. The current version of the manuscript is significantly improved in relation to the previous one. They corrected all aspects that were identified in the previous review. They did it in a right way and increased its scientific sound. Currently, the Reviewer positively assesses the paper and accept for its publication. Well done.